# TACTILE AFFORDANCE FOR ROBOT SYNESTHESIA FOR DEXTEROUS MANIPULATION

## ABSTRACT

In the field of dexterous robotic manipulation, integrating visual and tactile modalities to inform manipulation policies presents significant challenges, especially in noncontact scenarios where reliance on tactile perception can be inadequate. Visual affordance techniques currently offer effective manipulation-centric semantic priors focused on objects. However, most existing research is limited to using camera sensors and prior object information for affordance prediction. In this study, we introduce a unified framework called Tactile Affordance in Robot Synesthesia (TARS) for dexterous manipulation that employs robotic synesthesia through a unified point cloud representation. This framework harnesses the visuo-tactile affordance of objects, effectively merging comprehensive visual perception from external cameras with tactile feedback from local optical tactile sensors to handle tasks involving both contact and non-contact states. We simulated tactile perception in a virtual environment and trained task-oriented manipulation policies. Subsequently, we tested our approach on four distinct manipulation tasks, conducting extensive experiments to evaluate how different modules within our method optimize the performance of these manipulation policies.

## 1 INTRODUCTION

Tactile information is needed for robots to perform safe and precise manipulation in emerging domains like healthcare Erickson et al. (2018) and industrial assembly Tang et al. (2023). Recent years have seen a rapid growth in interest in vision-based tactile sensors like Gelsight Yuan et al. (2017a) and Punyo Alspach et al. (2019), which can provide high-resolution contact region observations by sensing the deformation internally via embedded cameras. Vision-based tactile sensors can be used to perceive objects' material properties Yuan et al. (2017b) and perform efficient manipulation She et al. (2021).

In human daily life, we perform fine motor actions through hand-eye coordination in a manner that requires almost no conscious effort. For instance, when our gaze captures an object, we use our hands to grasp it. During the grasping process, if the object is obscured by the hand, humans rely on tactile perception to grasp the object's immediate state. This close collaboration between visual and tactile feedback grants us the ability to manipulate objects dexterously, which is particularly crucial for the transition from coarse to fine task execution. However, for robots, naturally integrating visual and tactile modalities to accomplish operational tasks remains a significant challenge.

In the decision-making sequence of a manipulation task, tactile feedback is not always available. During these intervals, a robot can only rely on visual information to analyze the environment. However, when the robot's end-effector interacts with an object, visual information may be partially obscured, leading to the loss of critical data. The integration of visual and tactile information is particularly crucial for the precise manipulation of small objects. This dual reliance introduces two critical challenges: (i) the manipulation policy must effectively manage transitions between contact and noncontact states, and (ii) the policy must seamlessly integrate information from the inherently different visual and tactile modalities. Most existing research predominantly focuses on visual-tactile coordination in contact-rich scenarios or addresses visual and tactile information separately through distinct modules in tasks with limited contact.

In this paper, we present TARS, a framework designed to uniformly handle both contact and non-contact states while integrating visual and tactile modalities. Drawing on research in visual-tactile

synesthesia and visual affordances, we are the first to apply these concepts to a robotic system using optical tactile sensors and external cameras. We developed a unified point cloud visual-tactile processing module and a multi-state, multi-modal feature processing method trained through visual-tactile affordances. Additionally, we implemented a novel training-deployment framework based on the widely used Teacher-Student reinforcement learning framework for robotic tactile manipulation. Our framework can infer tactile affordances from visual input alone and supplement visual data with tactile information when available. This unified approach enables smooth transitions between contact and non-contact states, integrating visuo-tactile modalities to accomplish various manipulation tasks.

In our study, we used a widely adopted setup comprising of an external camera, a robotic arm, a two-finger parallel gripper, and an optical-based tactile sensor, which is prevalent in both academia and industry settings. We designed four manipulation tasks: Lift, Pick and Place, Pull Drawer, and Open Door. To increase the complexity, we restricted the completion of these tasks to the gripping actions of the two tactile sensors, making them more difficult than tasks without such restrictions. Additionally, unlike some studies that provide prior shape information, we relied solely on data from the external camera to emphasize generalization. In our ablation experiments, we validated the effectiveness of different modules within our framework and assessed its robustness under various physical conditions. Furthermore, we successfully conducted real-world experiments to demonstrate the applicability of our approach.

The rest of this paper is organized as follows: Section II presents related works, while the proposed TARS is detailed in Section III. In Section IV, we compare TARS with existing approaches through different manipulation experiments. Finally, Section V offers concluding remarks and outlines future work.

## 2 RELATED WORK

1) Visual-Tactile Coordination in Robotic Manipulation: Recently, there has been an increasing interest in designing novel tactile sensors Chi et al. (2018). Vision-based tactile sensors such as the Gelsight Yuan et al. (2017a), TacTip Ward-Cherrier et al. (2018) and DIGIT Lambeta et al. (2020) can provide high-resolution contact observations and have been proven useful in manipulation tasks Shah et al. (2021). Conversely, external RGB-D cameras provide critical global information but are prone to interference and occlusion. Some studies [9]–[13] have achieved visual-tactile modality fusion in contact-rich scenarios using sparse representations and self-attention methods. However, the sparse nature of tactile signals in many tasks limits these approaches' applicability. In low-interaction scenarios, image-based studies [14]–[17] have processed visual and tactile images separately using event cameras or gating mechanisms. These methods heavily rely on camera images when tactile signals are sparse, increasing the Sim2Real challenge. Additionally, research on visual-tactile coordination from a point cloud perspective [18], [19] has primarily focused on dexterous hands and force-tactile sensors, with limited exploration in diverse scenarios. In contrast, TARS builds on point cloud-based visual-tactile coordination methods to achieve a natural integration of visual and tactile modalities in robotic manipulation, as shown in Fig. 1. Our proposed framework incorporates robotic arms, parallel grippers, cameras, and optical tactile sensors. This framework facilitates seamless transitions between different contact states, leveraging both visual and tactile information to enhance robot performance, representing a significant advancement over existing approaches.

2) Visual-Tactile Affordance: Affordance is essential for robotic object manipulation, as it provides actionable information about how an object can be interacted with by robots. Several studies [20]–[23] have highlighted its effectiveness. For example, in point cloud-based robotic manipulation, [24] developed an end-to-end affordance method using reinforcement learning. Other works, such as [25]–[27], collected interaction data to pre-train affordance models before training manipulation policies based on these affordances. These methods, however, often require sampling surface point clouds from 3D CAD (Computer-Aided Design) models to obtain local contact points, relying on prior object information, which can be cumbersome. To address this issue, we conducted contact sampling on objects using both simulated and real optical tactile sensors to obtain precise local information. This approach simplifies the process of acquiring local points, making the affordance acquisition process independent of prior object information, thereby enhancing flexibility and applicability in various scenarios. 3) Point Cloud Based Visual-Tactile Synesthesia: In robotic manipulation research,

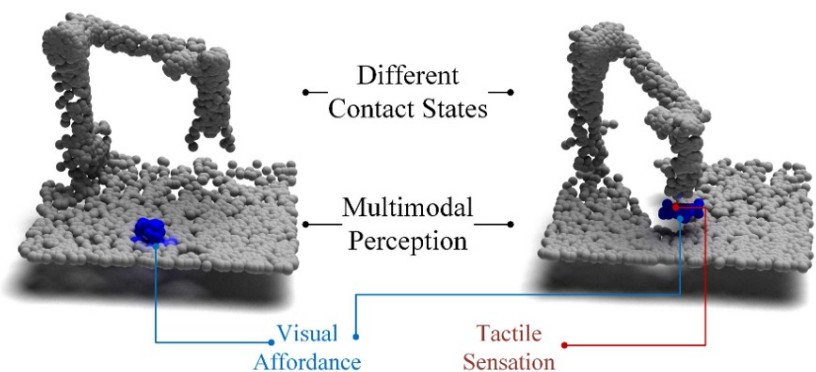

Figure 1: TARS (Tactile Affordance in Robot Synesthesia) provides sufficient information for manipulation tasks in both contact and non-contact states and under visuo-tactile multimodal conditions. We show different contact states during a grasping task.

some approaches [28], [29] rely solely on visual point clouds to enhance policy robustness, while others [30], [31] use tactile points to improve local perception. Studies on estimating object states [6], [32] often begin with a rough estimate using visual point clouds, refining the estimation with tactile point clouds for greater accuracy. This method encodes visual and tactile point clouds into a coherent 3D space, a concept known as robotic synesthesia [18], which has demonstrated strong capabilities in dexterous manipulation. However, these studies are generally limited to contact-rich states or in-hand manipulations. As shown in Fig. 2, our approach introduces visual-tactile synesthesia encoding based on optical tactile sensors, combined with visual-tactile affordance features to create a unified feature space. This method provides affordance perception through visual-tactile synesthesia in non-contact states and accurate visuo-tactile information in contact states, ensuring smooth transitions between these states. This integration enhances the continuity and effectiveness of manipulation policies across different interaction scenarios.

## 3 OUR APPROACH

We set up a robotic simulation environment in Isaac Gym and implemented tactile simulation using our method. Then, we use the Soft Actor-Critic (SAC) [33], a reinforcement learning algorithm, to train teacher policies for different tasks in the simulation environment with oracle observation. These policies are employed to train the Visual-Tactile Affordance (VTA) and Visual-Tactile Policy (VTP) modules within TARS. Finally, we deploy the trained VTA and VTP modules to realize robotic manipulation tasks.

### 3.1 SIMULATION OF TACTILE POINT CLOUD

The optical tactile sensor provides tactile information through images. By calibrating and modeling the sensor, we can extract simulated three-dimensional contact information from the two-dimensional tactile image data. In our framework, we decouple this tactile information by decomposing the three-dimensional contact information into a planar contact point and six-axis force information (Fig. 3). The six-dimensional contact force can be obtained in various ways through optical tactile sensors in real environments [34]–[36]. Building on these methods, we use tactile images from the real system as the input and employ a convolutional neural network (CNN) to predict sixdimensional contact forces. These predicted forces are then linearly adjusted to match the contact forces obtained in the simulations. By comparing the tactile images with reference images, we can obtain the planar contact points. In the real system, these points can be mapped to a contact point cloud through the calibration of the robotic arm's coordinate system with the camera's coordinate system. For the simulation environment, there are already many environments for tactile simulation [37]–[39], we aim for parallel training of our policy and choose Isaac Gym [40]. In our simulation, we modeled the contact scenario of Gelsight Mini [2] with a depth camera and force sensors to simulate contact states.

To represent visual and tactile data using a unified point cloud, we randomly sampled the simulated tactile depth images to obtain the contact point cloud.

## 3.2 VISUAL-TACTILE AFFORDANCE

The goal of the membrane model component is to establish a relationship between deformation of the bubble and their resulting forces. We model the bubble sensor as a homogeneous thin membrane, similar to Kuppuswamy et al. (2020). Our model considers three types of forces acting on each element of the bubble: Tension forces from neighboring elements, external forces from contact with the environment, and pressure force from the air inside. We consider the bubble deformation to be quasi-static, and solve for static equilibrium:

$$F_{tension} + F_{pressure} + F_{external} = 0. \tag{1}$$

We further linearize Equation 1 about a fixed reference configuration of the bubble, defined in subsection 3.1:

$$\delta F_{tension} + \delta F_{pressure} + F_{external} = 0, \tag{2}$$

where $\delta F_{external} = F_{external}$ since the reference configuration has no external forces acting on it.

We impose these equilibrium conditions at every mesh vertex and solve for values of these quantities at those vertices. For this discussion, $\delta F_i \in \mathbb{R}^{M \times 3}$ will denote a matrix of stacked node forces, for each of the three force quantities.

We first lump the change in pressure force to each vertex to compute $\delta F_{pressure}$. We lump the area to each vertex as $\vec{a} \in \mathbb{R}^M a_i = \frac{1}{3} \sum \text{area}(\Delta) \, \forall \, \Delta \in M i \in \Delta$. Similarly, we compute outward area-weighted vertex normals $\vec{n}_i \in \mathbb{R}^3$ for each vertex $i$. $\delta F_{pressure}$ can be computed as:

$$\delta F_{pressure} = \delta p a_1 \vec{n}_1^T a_2 \vec{n}_2^T \vdots a_M \vec{n}_M^T \tag{3}$$

where $\delta p$ is the difference between the observed pressure reading and the reference pressure reading.

$\delta F_{tension}$ is computed by assuming linear elasticity. We characterize the material by its Young's modulus $E$ and Poisson ratio $\nu$. Following Reissner-Minlin plate theory, we assume that the membrane thickness does not change during deformation and that the out of plane stress is zero (Hughes (2000 - 1987), Section 5.2). Additionally, we assume the bubble's bending stiffness is zero because the membrane is very thin (0.65mm) compared to its radius of curvature.

For a point on the mesh surface, we define the local coordinate system $ijk$ such that $i$ and $j$ are tangent to the surface and $k$ is perpendicular to the surface. (Refer to Figure 2.) Let $\sigma_{ab}$ refer to element $(a, b)$ of the stress tensor $\sigma$. $\varepsilon_{xx}$ refers to the normal strain in direction $x$, and $\gamma_{xy}$ refers to the shear strain in plane $xy$. From the frame of reference of such a point, the linear elasticity equations reduce to:

$$\varepsilon_{ii} = \frac{\sigma_{ii}}{E} - \nu \frac{\sigma_j}{E}, \qquad \varepsilon_{jj} = \frac{\sigma_{jj}}{E} - \nu \frac{\sigma_i}{E},$$
$$\gamma_{ij} = \frac{2(1 + \nu)}{E} \sigma_{ij}, \tag{4}$$

where the assumptions of no thickness change and no bending stiffness give additional constraints

$$\varepsilon_{kk} = \sigma_{kk} = \sigma_{jk} = \sigma_{ik} = 0. \tag{5}$$

These equations are applied on a triangular mesh, treating each triangle individually as a flat surface, following the procedure in Hughes (2000 - 1987) (Section 6.2.13).

As part of this simplification, when considering displacements of points in 3D, we project them onto the triangle plane for the purposes of calculating strain within each individual triangle. Let $B$ be the partial derivative matrix for a linear triangle element in 2D, such that

$$\varepsilon_{ii} \varepsilon_{jj} \gamma_{ij} = B \vec{u}_{o,2D}^T \vec{u}_{p,2D}^T \vec{u}_{q,2D}^T {}^T, \tag{6}$$

where $\vec{u}_{v,2D}$ refers to the 2D displacement of vertex $v$. For our simplification, we set

$$\vec{u}_{v,2D} = \text{project}(\vec{u}_v) = \hat{i}\hat{j}^T \vec{u}_v, \tag{7}$$

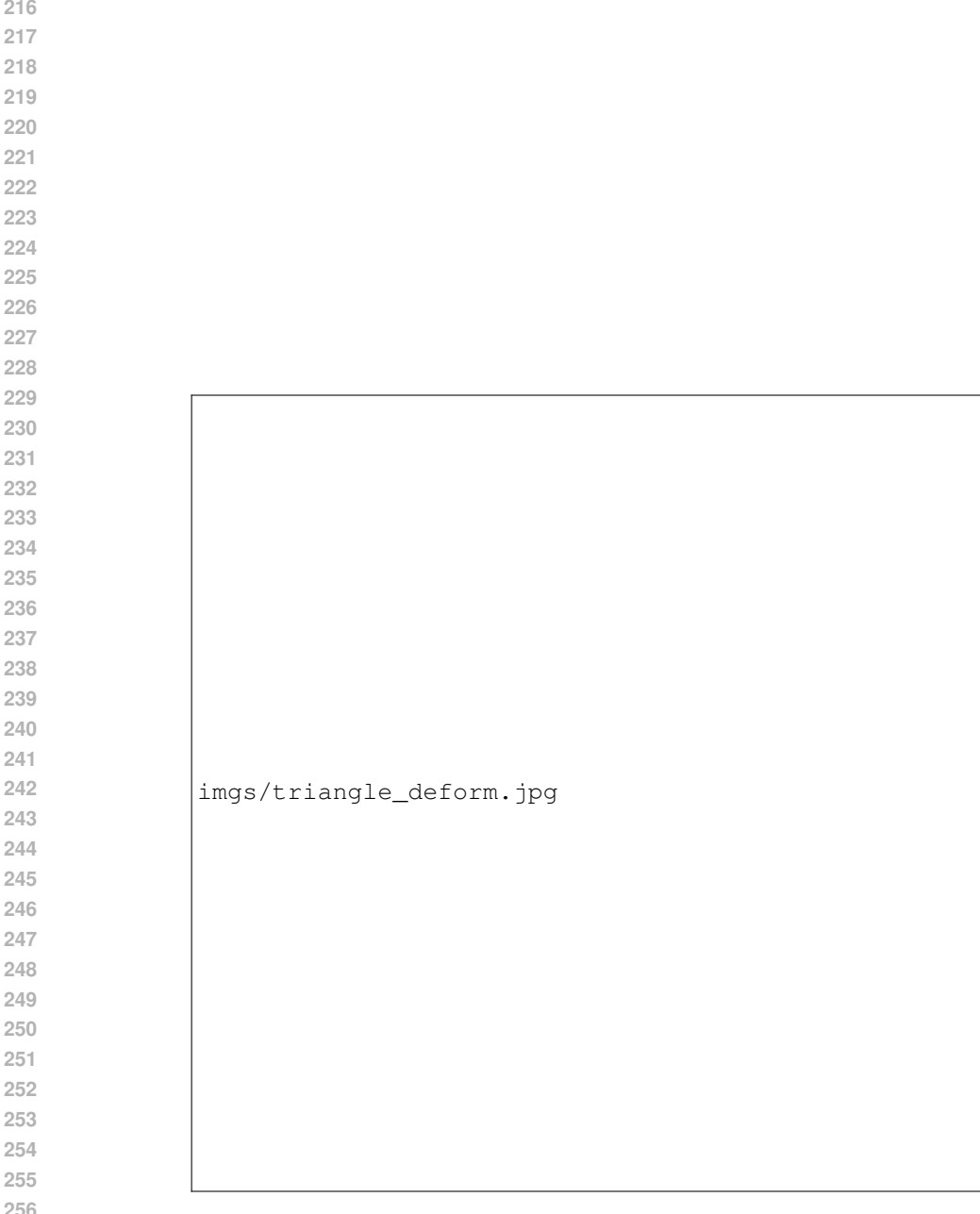

Figure 2: An illustration of the FEM setup. A local coordinate frame is computed for each triangle, and displacements are projected back to the local $ij$ plane to compute strains. [Best viewed in color]

where $\vec{u}_v$ refers to the observed 3D displacement of vertex $v$:

$$\vec{u}_v = \vec{x}_{v,cur} - \vec{x}_{v,ref} \tag{8}$$

Refer to Figure 2. The 2D computed node forces are translated back to 3D in an analogous fashion:

$$\vec{f}_v = \hat{i}\hat{j}\vec{f}_{v,2D}. \tag{9}$$

Following the standard FEM assembly procedure, we get a linear map $K$ between node displacements and node tension force changes:

$$\delta F_{tension} = K\vec{u}_1^T\vec{u}_2^T\ldots\vec{u}_{|M|}^T = K(\vec{U}) \tag{10}$$

where $\vec{U} \in \mathbb{R}^{3M}$ denotes a vector of stacked node displacements.

Substituting Equation 3 and Equation 10, in Equation 2, we obtain an expression for the unknown contact force $F_{external}$ as a function of the observed displacements $\vec{U}$ and pressure change $\delta p$:

$$F_{external}(\vec{U}, \delta p) = -K(\vec{U}) + \delta p a_1\vec{n}_1^T a_2\vec{n}_2^T \vdots a_M\vec{n}_M^T \tag{11}$$

To compute contact pressures $P \in \mathbb{R}^{M \times 3}$ at each node, we divide the computed node force by the area at each vertex:

$$P_{contact} = diag(\vec{a})^{-1}F_{external}. \tag{12}$$

We additionally define the continuous pressure distribution $\mathcal{P}_{contact}$ by barycentric interpolation of $P_{contact}$ within each triangular face.

The total contact force $\vec{f}_{net}$ is computed by summing the node forces across the mesh surface, excluding the boundary:

$$\vec{f}_{net} = \sum_{v \in M \setminus \partial M} F_{external\,v} \tag{13}$$

### 3.3 VISUAL-TACTILE POLICY

With the privileged information provided by the simulation environment, such as the position and pose of target objects, we can quickly obtain stable reinforcement learning policies through the parallelized simulation environment. However, this privileged information cannot be directly accessed in actual POMDP (Partially Observable Markov Decision Process) robotic systems. Therefore, we use a teacher-student learning approach to distill policies that can be applied in real-world environments from the trained models. The VTP (Visual-Tactile Policy) framework is illustrated in Fig. 4. We use the affordance trained by VTA and the visualtactile one-hot classification encoding together as point features, which ensures that our feature space is smooth. The point features have three dimensions: the first dimension is the affordance prediction ranging from 0 to 1, and the second and third dimensions represent the tactile and visual classification information. We will validate the roles of these features in Sec. IV-C. Through an encoder of PointNet, we can encode the coordinates and feature information of the point cloud into a feature vector. The student policy also employs a MLP as the decision network and we add a fully connected layer to output a Gaussian mixture density. We utilize a Gaussian Mixture Density Model (GMDM) to handle scenarios where multiple paths are planned for the same task by teacher policies. The final strategy samples one feasible path from the mixture density model. The loss function for the VTP module is shown as follows:

where i(a t |x) is a kernel function in the form of a multivariate Gaussian distribution with parameters μi ,i, a t represents the output action of teacher policy that should be taken, and x is the observation. The loss function (2) simultaneously trains the PointNet encoder network and the MLP policy network. The network models a probability density function (PDF) p(a s |x) as a mixture of m PDFs with the mixing coefficient = 0,1,...,m1. The student action as will be generated by sampling from p(a s |x). A parallelized teacher-student framework is then established for model training, incorporating our improvements. The VTP uses the policy trained by SAC as the teacher policy, while the DAgger [42] method mixes the decisions of the teacher and student policies. Additionally, a replay buffer was leveraged to utilize the data, continuously supplemented with new data through the parallelized environment of the Isaac Gym. In summary, our work establishes the synergistic using vision and

touch during manipulation processes. The pipeline of our framework is illustrated in Fig. 4, where our TARS framework comprises two key components: the VTA module, which provides affordance information, and the VTP module, which makes decisions using mixed encoding. Another significant aspect is our decoupling of tactile modality information to mitigate the transfer difficulty of the optical tactile sensor in sim-to-real scenarios. We decompose the tactile information provided by the optical tactile sensor into contact shape and contact force, and implement this method to achieve tactile perception in the simulation environment. This tactile decoupling approach enables the deployment of the VTA and VTP modules on real-world robotic systems.

## 4 EXPERIMENTS

The subsequent section evaluates TARS's performance in comparison to baselines and other variants in simulations. We focus on three key research questions: (1) How do visualtactile classification encoding and visual-tactile affordance contribute to policy performance? (2) How does the tactile point cloud influence grasping decisions? (3) Is our policy robust? These questions will be addressed in the following experiments.

### 4.1 EXPERIMENTAL SETUP AND TASKS DESCRIPTION

We evaluate our proposed method and comparison methods in the Isaac Gym physics simulator. In the simulation environment, we uniformly use the UR5 robotic arm and the Gelsight Mini tactile sensor simulation. We set the number of input points to 8192, including 128 tactile sampling points from two sensors in total, and tested this configuration across the four tasks. Additionally, we performed 4× point cloud downsampling, denoted as DS, and directly tested it on some tasks without altering the policy model. We selected single-stage tasks such as Lift Objects, Pull Drawer, and Open Door, as shown in Fig. 5. We guided the tasks through rewards to use the tactile sensors on the two-finger gripper to complete these tasks. Here is a brief introduction to these tasks: Lift Objects: There are irregular objects on the table with random initial positions and orientations. The agent needs to locate the object and identify its key parts, then use the tactile sensors to lift the object. Open Door: In the initial state, the door is closed. The agent needs to use the two tactile sensors on the parallel gripper to grasp the door handle and open the door to a specific angle. This task requires the agent to observe key positions of the door and achieve the task with a specific posture, making it very challenging. Pull Drawer: A drawer is initially closed, similar to open door, the agent needs to use the two tactile sensors on the parallel gripper to open the drawer to a specific distance. We also selected the multi-stage Pick and Place task for evaluation, as shown in Fig. 5: Pick and Place: An object with a random position and orientation is on the table. The agent needs to use tactile sensors to pick it up and place it at a target point on a separate, higher table. All the tasks mentioned above were trained using reinforcement learning with the oracle observation, resulting in high-success-rate teacher policies.

### 4.2 COMPARED METHODS

1) Baselines and Ablations: We compared our TARS with three main baselines. Our first baseline RS (Robot Synesthesia) refers to the SOTA (State of the Art) approach in [18], [19], where we use only the visual and tactile classification one-hot encoding for the features of the visual and tactile point clouds. In the second baseline VA (Visual Affordance), we did not use classification encoding for the visual and tactile point clouds; instead, we treat them uniformly as visual encoding and add our VTA module for prediction, referring to [24], [26]. In the third baseline PN+MLP (PointNet+Multilayer Perceptron), we retained only the positional features of the visual and tactile point clouds, setting other features to a uniform value [29]. The results of this approach will be further discussed in section IV-C. Additionally, following the approach in [24], we considered an end-to-end training method, where the policy network and the affordance network are trained simultaneously through the technique of reinforcement learning. However, we were unable to achieve successful convergence, so these results were not included in the comparisons.

2) Variants: We evaluated several variants of our model. For our method, we also tested its robustness under different settings. First, we examined whether the combined visualtactile perception maintained robustness with point clouds of varying scales. In the Lift task, we tested the applicability of the policy

trained on the Lightbulb object directly on other objects without modification. We also compared the impact of three different encoding inputs on the policy: our proposed TARS, the visual-tactile direct concatenation PN+MLP, and the PN+MLP without the tactile perception component. Additionally, we investigated the performance of policies based on different modalities during the training process of multi-stage pick and place tasks. This was done to validate the impact and contribution of our visual-tactile method on policy effectiveness.

### 4.3 SIMULATION RESULTS

1) Comparisons to Baselines: The comparison results, as shown in Tab. I, demonstrate that our method, which combines visuo-tactile classification encoding and visual affordance, achieves the best overall performance after extensive testing. In tasks involving rich contact, the RS method based on visuo-tactile classification encoding shows a significant improvement over the PN+MLP method. Similarly, in tasks with numerous non-contact states, the VA method based on visual affordance also demonstrates substantial improvement compared to the PN+MLP method. However, since noncontact scenarios are less frequent, the enhancement provided by VA is not as pronounced as that of RS. Additionally, our method shows its robustness to point cloud inputs of varying scales, indicating that the VTA module effectively learns the key tactile features of objects, enabling the policy to utilize this information effectively. 2) Variants: In the Lift task, we conducted direct tests without replacing the policy model. We selected six test objects out of twenty that were somewhat similar to the training object. We also used a visual point cloud policy as a baseline and evaluated the impact of removing the simulated tactile point cloud from the policy. The results, shown in Tab. II, indicate that our policy has strong generalization ability and that the local tactile perception significantly enhances the policy's performance. Among the test objects, the Apple produced anomalous results, likely due to its larger volume affecting the gripping policy across all three methods. In addition to evaluating the performance of the policy upon completion of the task training, we also measured the policy's performance at different training steps and recorded the results in Tab. III. The results indicate that during training, policies based solely on visual modality showed limited improvement after reaching a certain success rate. In contrast, visual affordance and tactile information were effective at different stages of training, with visual information aiding in the early stages and tactile information contributing in the later stages. This synergy resulted in our visual-tactile method achieving the best performance.

## 5 CONCLUSION

We presented a finite element force estimation method for soft-bubble grippers with only three parameters that can be calibrated with small amounts of data. Our model can run in near real-time and produce force predictions with accuracy beyond the current state of the art, especially for shear forces. In future work, we hope to develop a more accurate physical model for the bubble's deformation: Our current bubble model uses a simplified model of membrane deformations, ignoring any effect from changes in the bubble's curvature or from large displacements changing the orientation of individual mesh elements. Higher order elements including curvature effects should also improve accuracy. We also hope to achieve speed improvements by implementation in a compiled language.

## 6 ACKNOWLEDGEMENTS

This project has received funding from the National Key R&D Program of China (grant No.2022YFB4700400), National Natural Science Foundation of China (grant No.62073249), Key R&D Program of Hubei Province (grant No.2023BBB011).

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
