# OpenReview forum: "MimicTouch:Tactile Affordance for Robot Synesthesia for Dexterous Manipulation"
_ICLR.cc/2026/Conference — Submitted to ICLR 2026_

### Official Review · Reviewer_1qh1 · 2025-10-20

**Soundness:** 1
**Presentation:** 1
**Contribution:** 1
**Rating:** 0
**Confidence:** 5

**Summary:**

This paper is a duplicate submission of an already-published work: https://ieeexplore.ieee.org/document/10766612. The authors are also putting up a title from two irrelevant papers.

**Strengths:**

N/A

**Weaknesses:**

N/A

**Questions:**

N/A

**Details Of Ethics Concerns:**

This paper is a duplicate submission of an already-published work: https://ieeexplore.ieee.org/document/10766612. The authors are also putting up a title from two irrelevant papers.

---

### Official Review · Reviewer_yYCR · 2025-10-30

**Soundness:** 1
**Presentation:** 1
**Contribution:** 1
**Rating:** 0
**Confidence:** 5

**Summary:**

The paper is not ready

**Strengths:**

/

**Weaknesses:**

Minor weakness:

1.	Please use the same format for citation. Eg. The citation in introduction and in related works using different format in the text

2.	Figure 2 is missing

3.	Citations mentioned in the main text is missing in the reference section.

4.	“Our proposed framework incorporates robotic arms, parallel grippers, cameras, and optical tactile sensors. This framework facilitates seamless transitions between different contact states, leveraging both visual and tactile information to enhance robot performance, representing a significant advancement over existing approaches” Point cloud-based visual tactile fusion method is not novel. [1]

[1]Yuan et.al Robot Synesthesia: In-Hand Manipulation with Visuotactile Sensing

5.	The “3)” in line 107 is confusing, I do not find “1)” and “2)”

6.	Without literature review for recent state-of-art papers.

7.	No figure.4 in the paper but mentioned in the text.

Major weakness:
1.	Many Format Issues

2.	Writing and presentation is confusion and put barrier to reader.

3.	The major contribution claimed in line 53- 61 about “visual tactile synesthesia” and “teacher students” is not novel which has already been published in existing work [1], where they also use point cloud-based visual tactile fusion and teacher student reinforcement learning. The only difference might be the different sensor and the author using affordance in this paper.

4.	No tables and experiments results.

**Questions:**

See weaknesse above

**Details Of Ethics Concerns:**

The concept, learning pipeline and wording are close to existing paper: Robot Synesthesia:In-Hand Manipulation with Visuotactile Sensing

---

### Official Review · Reviewer_BB24 · 2025-11-01

**Soundness:** 1
**Presentation:** 1
**Contribution:** 1
**Rating:** 0
**Confidence:** 5

**Summary:**

The paper uses the latex template of ICLR 25 and the quality of paper is low in presentation, which seems like a draft with lot of information missing. For example, one of the figures is blank with caption and many figures referred in the main text are missing in the paper. Desk reject is suggested.

**Strengths:**

see the summary

**Weaknesses:**

see the summary

**Questions:**

see the summary

---

### Official Review · Reviewer_hREh · 2025-11-01

**Soundness:** 1
**Presentation:** 1
**Contribution:** 1
**Rating:** 0
**Confidence:** 5

**Summary:**

This paper is a copy of an existing paper "TARS: Tactile Affordance in Robot Synesthesia for Dexterous Manipulation" on RA-L. Additionally, this paper included an acknowledment, which violates double-blind policy.

**Strengths:**

N/A

**Weaknesses:**

N/A

**Questions:**

N/A

**Details Of Ethics Concerns:**

Copy-paste of existing paper and include the acknowledgement

---

### Meta-Review · Area_Chair_gnqv · 2026-01-02

**Summary:**

This paper proposes a visuotactile affordance framework for dexterous manipulation based on a unified point cloud representation. Reviewers consistently raise serious concerns about research integrity, originality, and submission quality, including the duplication of previously published work, violations of the double-blind policy, and substantial issues with presentation and completeness. These concerns remain unresolved, and the paper does not meet the basic standards for technical contribution or submission integrity. The AC recommends rejection.

**Reviewer Concerns:**

Reviewers identify fundamental issues that go beyond technical weaknesses. Multiple reviewers point out that the submission substantially overlaps with an already published paper, raising concerns about duplicate submission and research integrity. In addition, the paper exhibits severe problems in presentation and completeness, including missing figures, missing references, inconsistent formatting, and incomplete experimental results. Concerns are also raised about the lack of novelty relative to existing visuotactile fusion and robot synesthesia work. These issues are not addressed through the rebuttal process, and the concerns remain outstanding and blocking.

**Reviewer Scores:**

- Reviewer hREh: 0
- Reviewer BB24: 0
- Reviewer yYCR: 0
- Reviewer 1qh1: 0

---

### Decision · Program_Chairs · 2026-01-26

Reject